# Transcriptomic Classification of Pituitary Neuroendocrine Tumors Causing Acromegaly

**DOI:** 10.3390/cells11233846

**Published:** 2022-11-30

**Authors:** Julia Rymuza, Paulina Kober, Natalia Rusetska, Beata J. Mossakowska, Maria Maksymowicz, Aleksandra Nyc, Szymon Baluszek, Grzegorz Zieliński, Jacek Kunicki, Mateusz Bujko

**Affiliations:** 1Department of Molecular and Translational Oncology, Maria Sklodowska-Curie National Research Institute of Oncology, 02-781 Warsaw, Poland; 2Department of Cancer Pathomorphology, Maria Sklodowska-Curie National Research Institute of Oncology, 02-781 Warsaw, Poland; 3Department of Neurosurgery, Military Institute of Medicine, 04-141 Warsaw, Poland; 4Department of Neurosurgery, Maria Sklodowska-Curie National Research Institute of Oncology, 02-781 Warsaw, Poland

**Keywords:** neuroendocrine pituitary tumors, somatotropinoma, acromegaly, growth hormone, gene expression, transcriptome

## Abstract

Acromegaly results from growth hormone hypersecretion, predominantly caused by a somatotroph pituitary neuroendocrine tumor (PitNET). Acromegaly-causing tumors are histologically diverse. Our aim was to determine transcriptomic profiles of various somatotroph PitNETs and to evaluate clinical implication of differential gene expression. A total of 48 tumors were subjected to RNA sequencing, while expression of selected genes was assessed in 134 tumors with qRT-PCR. Whole-transcriptome analysis revealed three transcriptomic groups of somatotroph PitNETs. They differ in expression of numerous genes including those involved in growth hormone secretion and known prognostic genes. Transcriptomic subgroups can be distinguished by determining the expression of marker genes. Analysis of the entire cohort of patients confirmed differences between molecular subtypes of tumors. Transcriptomic group 1 includes ~20% of acromegaly patients with *GNAS* mutations-negative, mainly densely granulated tumors that co-express *GIPR* and *NR5A1* (SF-1). SF-1 expression was verified with immunohistochemistry. Transcriptomic group 2 tumors are the most common (46%) and include mainly *GNAS*-mutated, densely granulated somatotroph and mixed PitNETs. They have a smaller size and express favorable prognosis-related genes. Transcriptomic group 3 includes predominantly sparsely granulated somatotroph PitNETs with low *GNAS* mutations frequency causing ~35% of acromegaly. Ghrelin signaling is implicated in their pathogenesis. They have an unfavorable gene expression profile and higher invasive growth rate.

## 1. Introduction

Acromegaly is a severe and life-threatening disease caused by persistent excess of growth hormone (GH), which stimulates synthesis and secretion of the insulin-like growth factor-1 (IGF-1). In the majority (95%) of patients, acromegaly is caused by sporadic GH-secreting pituitary neuroendocrine tumor (PitNET). High IGF-1 level promotes cell proliferation, inhibits apoptosis, and causes most of the clinical symptoms of acromegaly, ranging from subtle to severe, including limb hypertrophy, soft tissue edema, arthralgia, prognathism and hyperhidrosis to frontal bone hypertrophy, diabetes mellitus, hypertension, and respiratory and heart failure. Excessive body growth and gigantism may develop when a somatotroph tumor develops in young patients before closing the epiphyses of long bones [1].

Acromegaly is a particular disease entity that is generally treated with somatostatin analogs and surgery. However, PitNETs that cause clinical symptoms are quite heterogenous, varying in terms of pathomorphological characteristics, radiological imaging results, invasiveness, and molecular features [2,3]. Most of somatotroph tumors express GH and PIT-1 transcription factor and are further divided into sparsely and densely granulated (SG and DG, respectively) with electron microscopy-based evaluation or anti-cytokeratin staining. These two types of somatotroph tumors differ in magnetic resonance imaging (MRI) features and clinical course of the disease [4]. Acromegaly can also be caused by mammosomatotroph and mature plurihormonal tumors, characterized by expression of GH and additional hormones (prolactin (PRL) or PRL and thyroid-stimulating hormone (TSH), respectively). Additionally, it may be caused by mixed somatotroph–lactotroph tumors (composed of two distinct populations of somatotroph and lactotroph tumor cells), poorly-differentiated Pit1-lineage tumors, and acidophil stem cell tumors features [2,3]. SG somatotroph PitNETs and plurihormonal PIT-1 positive PitNETs are categorized as high-risk tumors due to characteristics of aggressiveness including invasive growth and higher recurrence rate [5].

The most common molecular changes in somatotroph tumors are activating mutations in *GNAS* gene, which encodes a stimulatory subunit of heterotrimeric G protein. These mutations are present in approximately 40% of somatotroph PitNETs [6]. They cause hyperactivation of cAMP-dependent pathways and, consequently, both increased secretion of GH and proliferation of somatotropic cells [7]. Clinical significance of *GNAS* mutations is unclear; however, they are certainly more common in DG than in SG tumors [8]. 

Recently, a subset of somatotroph PitNETs with elevated expression of gastric inhibitory polypeptide receptor (GIPR) was identified [9,10]. In these tumors *GIPR* expression is considered to additionally stimulate cAMP pathway causing paradoxical GH response to glucose intake [11]. These tumors were suggested to constitute a separate molecular subgroup as they differ in epigenetic profile and high *GIPR* expression is mutually exclusive with *GNAS* mutations [12].

Considering the complex nature of the clinical and pathological spectrum of PitNETs causing acromegaly we aimed to investigate gene expression in somatotroph tumors to verify whether transcriptomic profiles correspond to current histological/clinical classification.

## 2. Materials and Methods

### 2.1. Patients and Tissue Samples

This study included 134 patients with biochemically confirmed acromegaly that were treated with transsphenoidal surgery in two specialized centers: the Department of Neurosurgery, Military Institute of Medicine, Warsaw and the Department of Neurosurgery, Maria Sklodowska-Curie National Research Institute of Oncology, Warsaw in the years 2013–2020. Diagnostic criteria of acromegaly based on clinical characteristics, increased serum IGF-1 levels, and non-suppressible GH after oral glucose tolerance test (OGTT) in patients with no OGTT contraindications were used. All patients received somatostatin receptors ligands (SRLs) treatment (octreotide or lanreotide) before surgery following the recommendations of Polish Society of Endocrinology [13]. Invasive growth was determined based on preoperative MRI using Knosp classification. Tumors scored with Knosp grades 0–2 were considered noninvasive, while those with Knosp grade 3 and 4 were considered invasive [14].

Each tumor sample was divided in three parts. One of them was snap frozen in liquid nitrogen and stored for molecular analysis and the other two were preserved for histopathological evaluation, including immunohistochemical staining and ultrastructural analysis with electron microscopy. Pathomorphological diagnosis was based on evaluation of the immunoexpression of pituitary hormones (GH, PRL, ACTH, TSH, FSH, LH, α-subunit) and Ki-67, as well as assessment of ultrastructural status (sparsely vs. densely granulated tumors). Expression of PIT-1 transcription factor was confirmed with immunohistochemical staining retrospectively since a large proportion of the tumors were originally diagnosed with WHO 2014 criteria that did not comprise evaluation of transcription factors. Overall patients’ characteristics are presented in Table 1.

The study was approved by the local Ethics Committee of Maria Sklodowska-Curie National Research Institute of Oncology in Warsaw, Poland. Each patient provided informed consent for the use of tissue samples for scientific purposes. 

DNA and total RNA from tumor samples were isolated with AllPrep DNA/RNA/miRNA Universal Kit (QIAGEN) and stored at −70 °C.

### 2.2. Testing for GNAS Mutation Status

The presence of *GNAS* point mutation (exons 7 and 8) was assessed with Sanger sequencing in 134 tumor samples. DNA was amplified in PCR with FastStart Taq DNA Polymerase (Roche Diagnostics, Mannheim, Germany) using GeneAmp 9700 PCR system (Applied Biosystems, Foster City, CA, USA). PCR product was purified with ExoStar (GE Healthcare Life Sciences, Pittsburgh, PA, USA), labeled with BigDye Terminator v.3.1 (Applied Biosystems), and analyzed by capillary electrophoresis using ABI PRISM 3300 Genetic Analyzer (Applied Biosystems). PCR primers’ sequences are provided in Appendix A.

### 2.3. RNA Sequencing 

A total of 48 tumor samples were subjected to RNA sequencing (RNA-seq). We included pure somatotroph PitNETs (mixed GH/PRL, mammosomatotroph, or plurihormonal tumors were not included) with equal proportions of SD and DG tumors. Library preparation was performed with 1 μg RNA from each sample using NEBNext Ultra II Directional RNA Library Prep Kit for Illumina. NEBNext rRNA Depletion Kit was used for ribosomal depletion. The quality of libraries was assessed using the Agilent Bioanalyzer 2100 system (Agilent Technologies, Santa Clara CA, USA). Libraries were then sequenced on an Illumina NovaSeq 6000 platform, and 150-bp paired-end reads were generated. A minimum of 30 M read pairs per sample were generated. Sequencing was performed by Eurofins Genomics service.

### 2.4. Analysis of RNA-Seq Results

Quality control of raw reads was conducted using FastQC [15]. Raw reads were mapped to the human reference genome GRCh37/hg19 with HISAT2 [16]. The raw unnormalized count matrix was generated using featureCounts [17] with gene features from GENCODE (v39) and imported to DESeq2 [18]. Low-expression genes (genes with less than five sequencing reads in less than 25% of samples) were excluded from further analysis. Filtered matrix was normalized using DESeq2 [18] and used for sample clustering with k-means algorithm (R package cluster [19]) and hierarchical clustering (Manhattan distance and ward.D agglomeration, R library stats). Analysis of genes differentially expressed between clusters (transcriptomic groups) was performed using DESeq2 [18]. Differentially expressed genes (DEGs) were defined as those with adjusted *p*-value < 0.05 and fold change value (│FC│) > 2. Gene set enrichment (GSE) analysis was conducted with fgsea [20]. Additionally, marker genes for each group were detected using R package MGFR.

### 2.5. Quantitative Reverse Transcription PCR (qRT-PCR) Gene Expression Analysis

One microgram of RNA was subjected to reverse transcription with Transcriptor First Strand cDNA Synthesis Kit (Roche Diagnostics, Mannheim, Germany). qRT-PCR reaction was carried out in 384-well format using 7900HT Fast Real-Time PCR System (Applied Biosystems, Foster City, CA, USA) and Power SYBR Green PCR Master Mix (Thermo Fisher Scientific, Waltham, MA, USA) in a volume of 5 μL, containing 2.25 pmol of each primer. The samples were amplified in triplicates. *GAPDH* and *SDHA* served as reference genes. Delta Ct method was used to calculate relative expression level with geometric mean of reference genes Ct value for normalization. PCR primers’ sequences are presented in Appendix A.

### 2.6. Immunohistochemical Staining

Immunohistochemical staining (IHC) was performed on 4 μm FFPE tumor sections with the use of Envision Detection System (no. K500711-2, DAKO, Glostrup, Denmark). Tissue samples were deparaffinized with xylene and rehydrated in a series of ethanol solutions of decreasing concentration. Heat-induced epitope retrieval was carried out in a Target Retrieval Solution pH 9 (DAKO) in a 96 °C water bath, for 30 min. Tumor sections were treated with a blocker of endogenous peroxidase (DAKO) for 5 min and then incubated for 1 h with the primary antibody anti-Pit-1 (clone D-7; dilution 1:2000) (sc-393943; Santa Cruz Biotechnology, Dallas, TX, USA) or ant-SF1 (clone A1; dilution 1:500) (sc-393592; Santa Cruz Biotechnology). The color reaction product was developed with 3,3′-diaminobenzidine tetrahydrochloride (DAKO) as a substrate; hematoxylin counterstaining was applied for nuclear contrast.

### 2.7. Statistical Analysis and Data Visualization

Two-sided Mann–Whitney U-test was used for analysis of continuous variables. The Spearman correlation method was used for correlation analysis. Significance threshold of α = 0.05 was adopted. Data were analyzed and visualized using GraphPad Prism 6.07 (GraphPad Software) and R environment. R libraries such a as ggplot2 [21] and plotly [22] were used for visualization. Moreover, scaled normalized RNA-seq read counts were visualized on KEGG pathway graph using pathview [23]. In pathview visualization, the median of scaled normalized RNA-seq read counts for multiple genes were presented for the pathway elements if it was composed of more than one gene, according to KEGG Pathway Database annotation. Hsa04935 KEGG pathway was used.

## 3. Results

### 3.1. Incidence of GNAS Mutations

First, we determined *GNAS* mutational status in tumor samples in the entire cohort of patients. The missense mutations were identified in 52/134 (38.8%) patients. A total of 40 patients harbored mutations in exon 8 including 35 variants c.C601T:p.R201C, 4 c.A680G:p.Q227R, and 2 c.C601A:p.R201S. A total of 12 patients had mutations in exon 9 including seven variants c.A680AT:p.Q227L and five c.A680G:p.Q227R. We did not observe significant relationship between *GNAS* mutation and demographical/clinical features including patients’ age, gender, pathological diagnosis, invasiveness status, and tumor size.

### 3.2. Whole Transcriptome Analysis

A total of 48 tumor samples were successfully processed. An average of 87,347,293 reads per sample were generated with an average 90.79% reads mapped to UCSC hg19 reference genome. The sequencing reads were mapped to 19,631 human protein-coding genes, and 16,096 mapped protein-coding genes remained for inclusion in subsequent analyses after low-expression genes were filtered out.

Data-dimensionality reduction analysis including principal component analysis (PCA) and uniform manifold approximation and projection (UMAP) clearly indicated the presence of three separate transcriptomic groups of tumor samples with groups 1 and 2 being more similar to each other than to the third group (Figure 1A). The same pattern was observed in hierarchical clustering analysis where three basic branches of clustering tree corresponded to transcriptomic groups observed in PCA and UMAP results (Figure 1B). Nearly the same clustering results were observed regardless of the number of differentially expressed genes included in the analysis. The results for 1%, 10%, and 20% of most differentially expressed genes are presented in Appendix A. The preliminary analysis of clinical data for discovery set of 48 samples revealed notable differences between three identified transcriptomic groups. Group 1 included *GNAS* wild-type (*GNAS*wt) DG tumors (except for one *GNAS*wt, SG tumor), group 2 included mainly DG tumors with a high proportion of samples with *GNAS* mutation, and group 3 included basically SG tumors with a low percentage of *GNAS*-mutated (*GNAS*mut) ones. 

We determined the genes that are differentially expressed between each of the transcriptomic groups by comparing each group with the remaining groups separately (group 1 vs. group 2, group 2 vs. group 3, and group 2 vs. group 3). We found that paired groups differ in the expression of high number of genes. Specifically, 1007 differentially expressed genes (DEGs) that met criteria │FC│ > 2 and adjusted *p*-value < 0.05 were found when comparing groups 1 and 2, 2403 DEGs were identified when comparing groups 1 and 3, and 1685 DEGs were found in comparison of groups 2 and 3. Results of differential analysis are presented in Figure 1B. The lists of differentially expressed genes are reported in Appendix A.

The functional implications of the difference in gene expression were investigated with GSE analysis with Gene Ontology (GO) including GO Biological Processes and GO Molecular function. The analysis resulted in the identification of a large number of significantly enriched GO terms. The most significantly enriched GO Biological Processes, according to the highest significance level were the terms related to G-protein signaling, ion transport, cellular adhesion, and differentiation, while the most enriched GO Molecular function terms were those related to signaling and ion transport. The top 10 most enriched terms for each comparison are presented in Figure 2; all significantly enriched terms are listed in Appendix A.

### 3.3. Differences in the Expression of Genes Involved in GH Secretion Pathway

With a given high number of DEGs we paid a special attention to those that are related to GH secretion. According to literature data growth hormone secretion is primarily induced by hypothalamic somatoliberin (GHRH) as well as by ghrelin and GIP through activation of the corresponding membrane receptor on somatotroph cells. Interestingly, we found that the three identified transcriptomic groups of somatotroph PitNETs differ significantly in the expression of genes coding for each receptor. Tumors within groups 1 and 2 have high expression of *GHRHR* (somatoliberin receptor), but lower expression of *GHSR* (ghrelin receptor) than tumors in group 3. Tumors in group 1, in addition to high *GHRHR* expression, have notably higher expression of *GIPR* than the two remaining transcriptomic groups. Differences were also found among genes of somatostatin and dopamine receptors that play a role in modulation of somatotroph secretory activity. Expression of *SSTR3* was at a notably lower level in group 2, while a significant decrease of *SSTR5* in group 1 was observed as compared to groups 2 and 3. Higher expression of dopamine receptors *DRD1* and *DRD2* was in turn found in group 3. Irrespectively to receptor activation, Ca^2+^ influx through voltage-gated Ca^2+^ channels (VGCCs) also plays an important role in GH secretion [24]. *CACNA1C* and *CACNA1D* encoding VGCCs were identified as expressed at significantly higher levels in groups 1 and 2 as compared to group 3 (Figure 2). The signaling pathways downstream key receptors (cAMP and Phospholipase C pathways) are mediated by many proteins that are encoded by the genes that were differentially expressed between three transcriptomic groups. They include genes encoding G-proteins: *GNAI1*, *GNAI2*, adenyl cyclase *ADCY1*, *ADCY3*, *ADCY4*, *ADCY5*, *ADCY7*; cAMP response elements *CREB1*, *CREB3L1*, *ATF2*, *ATF5B*, *ATF4*, *CREB*, *CREBP*, *EP300*; phospholipases C: *PLCB2*, *PLCB4*, *PLCB1*; and protein kinases C: *PRKCA*, *PRKCD*, *PRKCE*, *PRKCI*, *PRKCZ,* as well as inositol trisphosphate receptor *ITPR3*. Details are presented in Appendix A, while comprehensive representation of the expression levels of genes involved in GH secretion is presented in Figure 2, which is based on KEGG pathway (visualized with pathview, original picture available in Appendix A). *GH1* encoding GH was also found differentially expressed, with the highest expression in transcriptomic group 1 and the lowest in group 3 (Figure 3A).

### 3.4. Differences in the Expression of Known Genes Involved in Somatotropinoma Clinical Outcome

Aberrant gene expression was previously shown to be involved in acromegaly patients’ outcome and response to SRLs. Beside the role of somatostatin receptors, the role of genes involved in epithelial-mesenchymal transition, cell proliferation, and cell signaling was previously reported [26]. Transcriptomic groups of somatotroph PitNETs differed in the expression of genes related to epithelial-mesenchymal transition (EMT) that have a proven role in acromegaly, including *CDH1* [27,28], *SNAI2* [29], *FLNA* [30], *ARRB1* [31,32], *RORC* [33], and *ESRP1* [28], but also in other genes with an important role in EMT including *CDH2*, *CDH3*, *CDH11*, *CTNNB1*, *CLDN1*, *CLDN3*, *CLDN4*, *CLDN9*, and *ZAEB1* (Figure 4A). Differences between transcriptomic groups were also observed in expression levels of proliferation-related genes *CCND1* [34], *CDKN1B* [27], and *MKI67* [35] and genes involved in cell signaling *TGFB1* [36] and *STAT3* [37] that all have a reported role in acromegaly patients’ outcome (Figure 4B).

### 3.5. Role of SF-1 (NR5A1) Transcription Factor in a Subtype of Somatotroph Tumors

We explored the expression levels of known genes encoding transcription factors specific to particular lineages of anterior pituitary cells in three identified transcriptomic groups of somatotroph tumors. A striking difference in the expression level of *NR5A1* (SF-1) was observed. It is expressed at very high level in tumors from group 1 as compared to other tumors (Figure 5A). SF-1 (*NR5A1*) is a commonly accepted marker of pituitary gonadotroph cell lineage; therefore, its higher expression in group 1 of somatotropinomas requires special attention. We measured *NR5A1* expression in somatotroph tumors of transcriptomic group 1 and gonadotroph PitNET samples (tumor samples from our previous investigation [38]) with qRT-PCR. This analysis showed that the range of *NR5A1* expression in group 1 of somatotroph PitNETs and in gonadotroph PitNETs is similar (Figure 5B). Neither *POU1F1* nor *TBX19* were among the genes differentially expressed between tumors forming the three transcriptomic groups (Figure 5A).

Transcriptomic group 1 tumors are those with high *GIPR* expression. GIPR was previously found to be involved in steroidogenesis process. It induces expression of known steroidogenesis-related genes including *NR5A1*, *STAR*, and *CYP11A1* [39,40]. As expected, higher expression of these genes was found in group 1 tumors as compared to each of the other groups. Accordingly, a significant correlation of the expression levels of *GIPR* and each of *NR5A1*, *STAR,* and *CYP11A1* was found, with the highest correlation coefficient in *GIPR* and *NR5A1* analysis (Spearman R = 0.785, *p* < 0.0001) (Figure 5C). 

Next, we determined whether tumors of transcriptomic group 1 that express *NR5A1* are positive for protein SF-1 expression as determined by immunohistochemical staining. As a result, each of the samples showed clear nuclear immuno-reactivity with antibodies against SF-1 over the entire tumor tissue sample area. The representative examples of the results of staining for PIT-1 and SF-1 are presented in Figure 5D.

### 3.6. Difference in Clinical/Histopathological Features between Transcriptomic Groups of Somatotroph Tumors

To evaluate the differences in clinical parameters between the transcriptomic groups of somatotroph tumors we used tumor samples from the entire cohort of 134 acromegaly patients, without any intentional preselection. Unfortunately, in our study we were unable to include more than 48 RNA samples in RNA-seq procedure. Therefore, we determined whether somatotroph tumor samples can be classified and assigned to particular transcriptomic groups by qRT-PCR-based evaluation of expression level of the marker genes. Using RNA-seq results we selected nine genes that could serve as potential classifiers. We measured the expression level of nine potential markers with qRT-PCR in 48 samples that were previously included in transcriptome profiling and that were clearly categorized (Figure 6A). Using receiver operating characteristic (ROC) curve analysis we determined the value of qRT-PCR-measured expression levels as classifiers of each transcriptomic group. The results of the evaluation of the nine marker genes are presented in Appendix A. We selected three marker genes (*NR5A1*, *CCND2,* and *SEC23A*) that met criteria of area under curve (AUC) > 0.99 and that allowed for selecting a clear threshold value. *NR5A1* distinguished between transcriptomic group 1 and groups 2/3, while *CCND2* and *SEC23A* discriminated groups 1/2 and group 3 (Figure 6B). The use of thresholds for *NR5A1*, *CCND2,* and *SEC23A* expression level determined by data from the RNA-seq group allows for clear categorization of nearly the entire patient cohort (Figure 6C). Six patients (4.5%) of the entire patient cohort could not be assigned to any transcriptomic category due to a low expression level of each marker. These six patients were excluded from further analysis of clinical data.

The analysis of clinical data showed that in transcriptomic group 1, composed of tumors positive for *NR5A1* expression, there are no mutations of *GNAS*. This group includes 25/128 patients (19.5%). Most of the tumors within this group were DG somatotroph PitNETs, however, five of the tumors were positive for both GH and LH upon immunohistochemical staining and were classified as plurihormonal GH/LH tumors. Transcriptomic group 2 accounted for 59/128 patients (46.1%). The vast majority of group 2 patients (66.1%) had tumors with mutations in *GNAS* gene, mainly determined as densely granulated. This group included DG somatotroph PitNETs, but also mixed GH/PRL tumors. Transcriptomic group 3 included 44/128 patients (34.4%) with low incidence of *GNAS* mutations (18% were *GNAS*mut). The difference in proportions of *GNAS*mut patients between transcriptomic groups was significant (Chi square test *p* < 0.0001). Group 3 was composed mostly of sparsely granulated tumors as determined with electron microscopy, diagnosed mainly as SG somatotroph PitNETs. It also included seven DG tumors (10%), but no mixed or plurinominal PitNETs. In this group, a higher proportion of invasive tumors (43%, 19/44) was observed, as compared to transcriptomic groups 1 and 2 (16.6% and 20.3%, respectively) (Chi square test *p* = 0.0154). Results are presented in Figure 7A. Tumors from transcriptomic group 2 were significantly smaller than those from two other groups (Figure 7B). No differences were observed in serum GH and IGF-1 levels (Figure 7B) or patients’ demographical data (age and gender) between the groups.

RNA-seq results revealed that transcriptomic groups differ in the expression level of key genes involved in GH synthesis and secretion including *GHRHR*, *GIPR*, *GHSR*, *SSTR*, and *DRD1* as well as *GH1* encoding GH itself. Using qRT-PCR we measured expression level of these genes in the entire patient cohort categorized into three transcriptomic groups according to marker genes evaluation. Significant differences in the expression of each gene were found and confirmed the observation in whole transcriptome analysis. Groups 1 and 2 have high levels of *GH1* and *GHRHR*. Tumors in group 1 have high *GIPR* expression, while those in group 3 have higher expression of ghrelin receptor (*GHSR*) and *DRD1*. A slight discrepancy was observed between qRT-PCR results of the entire cohort and data from RNA-seq in terms of *SSTR5* expression level. PCR-based measurement showed the highest level of *SSTR5* expression in transcriptomic group 3. Results are visualized in Figure 8. 

## 4. Discussion

The symptoms of acromegaly result from GH oversecretion by pituitary tumors which develop from PIT-1-positive anterior pituitary cell lineage. Our results show that the profile of gene expression clearly discriminates somatotroph tumors into three transcriptomic groups that differ in the expression level of a large number of genes. According to GSE results, the expression differences are mainly related to process of cell signaling including G-protein signaling and ion transport, both of which play key roles in regulation of secretory activity of pituitary cells [7,41] as well as to cellular adhesion and differentiation.

Importantly, DEGs include genes directly involved in regulation of GH secretion like those coding for receptors of somatoliberin, gastric inhibitory polypeptide and ghrelin, and receptors for dopamine and somatostatin, as well as voltage-gated calcium channels. It seems that secretory activity in tumors of each transcriptomic subgroup may be driven by a slightly different mechanism. The first group is characterized by high expression of *GHRHR*, *GIPR*, and *VGCCs* as well as the highest expression of GH-encoding gene. The second group includes tumors which seem to be mainly related to hypothalamic GHRH stimulation, as they show high *GHRHR* expression. Tumors in the third group, in turn, are probably dependent on ghrelin signaling since they present with the highest expression of gene encoding ghrelin receptor.

*GIPR*, which is highly expressed in group 1 tumors, encodes the receptor of glucose-dependent insulinotropic polypeptide (GIP) that has a well-known role in neuroendocrine tumors [25]. Upon ligand binding GIPR activates coupled heterotrimeric G-protein complex containing a stimulatory G subunit and leads to activation cAMP pathway. Therefore, GIPR activation is considered to be mimicking GHRHR stimulation by hypothalamic GHRH hormone [25]. Acromegaly patients with GIPR expression commonly react to oral glucose load with a paradoxical increase in GH level, which resembles the induction of hormone secretion in food-dependent Cushing’s syndrome [11]. DNA methylation profiling in *GIPR*-high versus *GIPR*-low somatotroph PitNETs showed a difference in genome-wide methylation pattern which is in line with our results, indicating that this group represents a separate molecular subtype [12]. Interestingly, high expression of *NR5A1* encoding SF-1 transcription factor was found exclusively within this transcriptional group. According to classification of PitNETs, SF-1 is a well-established marker of gonadotroph tumors and the observation that it is also expressed in a particular subtype of somatotroph tumors may suggest a need of slight revision of classification criteria. We observed that *GIPR*-high somatotroph PitNETs express *NR5A1* at the level comparable to gonadotroph tumors. Of note, the expression of *NR5A1* in some somatotroph PitNETs was also recently noticed by Mario Neou et al. [42].

The results of our immunohistochemical staining with antibodies against SF-1 and PIT-1 in tumors of transcriptomic subtype 1 clearly show that they are expression-positive for both transcription factors. Nuclear SF-1 immunoreactivity was observed in these tumors, as could be expected in staining of gonadotroph PitNETs according to the most up-to-date diagnostic criteria [43]. The expression of both PIT-1 and SF-1 was previously shown in rare double pituitary adenomas that are composed of multiple PitNETs in one tumor [43,44]. In double PitNETs two areas of tumor tissue with distinct expression of PIT-1 or SF-1 can be observed [44]. In contrary to double tumors our tissue staining of somatotroph PitNETs of transcriptomic group 1 clearly showed the co-expression of PIT-1 and SF-1 over the entire tumor sample area, clearly indicating the homogenous nature of these double-positive somatotroph PitNETs. From a histological point of view, they are unequivocally somatotroph tumors (based on both immunohistochemistry and electron microscopy) [45].

In our study we clearly demonstrated that somatotroph tumors with SF-1 expression are those that express *GIPR*. It appears to be functionally related. *GIPR* plays a role in steroidogenesis, as previously observed in adrenal cortex-derived cell line and mice. The experiments showed that manipulating *GIPR* expression results in changes of transcription levels of *NR5A1*, *STAR,* and *CYP11A1* [39,40]. Transcriptomic group 1 tumors are those with significantly highest expression of *GIPR*, *NR5A1*, *STAR*, and *CYP11A1* as compared to tumors from other groups. Accordingly, correlation analysis indicates that these genes are co-expressed with *GIPR*. This suggests that high expression level of *NR5A1* in transcriptomic group 1 of somatotroph tumors results from high GIPR-related signaling. The mechanism underlying high *GIPR* expression in somatotroph tumors was the matter of previous research and it is still unclear [12].

High expression of *NR5A1* in tumors in transcriptomic group 1 may suggest the potential contamination of tissue samples with normal pituitary tissue. In fact, subtype 1 tumors do express *NR5A1*, but they do not differ in the expression of other transcription factors specific for pituitary lineages, such as *TBX19*. If our results would be biased by the presence of normal anterior pituitary, we would expect the difference in the expression of other markers of pituitary lineages. The clearly observed co-expression of PIT-1 and SF-1 in this subtype of somatotroph PitNETs (Figure 5D) also shows that this result of transcriptome-based classification is not biased by normal pituitary. We cannot exclude any potential contamination with normal tissue at sampling procedure but we are convinced that it did not significantly affect the results of our study.

Transcriptomic classification of acromegaly-causing tumors may have clinical implications. The use of SRLs is basic pharmacological treatment for acromegaly patients. Therapy response and prognosis were previously found to be related to expression of somatostatin receptor genes (*SSTR2* and *SSTR5*) [27,46], genes involved in EMT [26], and those involved in cell proliferation like *CDKN1B* (p27) [27] or *MKI67* (Ki-67) [35]. Loss of cadherin E was clearly linked with a poor response to SRLs [47]. Importantly, striking differences in the expression of *CDH1* were observed between transcriptomic groups with the highest expression in group 2 and the lowest in group 3. The expression profile of other genes that were previously shown to be related to SRLs response [27,28,31,33,46] suggests also that tumors in transcriptomic group 2 may be the most sensitive to somatostatin analogs. They have a favorable pattern of gene expression—those related to EMT (high *RORC* and *ESRP1* and low *ARRB1* (β-arrestin)) as well as *SSTR5* (the highest expression among subtypes), *MKI67* (low expression), and *CDKN1B* (high level).

Preliminary analysis of clinical data from samples included in RNA-seq showed that transcriptomic profile corresponds to pathomorphological diagnosis. In general, transcriptomic group 1 includes *GNAS*wt, densely granulated somatotroph PitNETs, group 2 includes *GNAS*wt and *GNAS*mut DG tumors, and transcriptomic group 3 includes predominantly SG tumors. Unfortunately, the proportions between the pathomorphological subtypes of somatotroph tumors among the samples included in our RNA-seq do not reflect those in general population of acromegaly patients. Therefore, we made an attempt to evaluate the true proportions in patient numbers in individual groups and clinical significance of transcriptomic classification in a large group of acromegaly patients, without any intentional preselection. We evaluated candidate marker genes that allow for stratification of patients into three transcriptomic subtypes and used three most potent markers to classify the entire patient cohort. This classification based on three markers was not adequate only in six patients that were subsequently excluded from the analysis. The results allow some general conclusions to be drawn. They clearly show that in transcriptomic group 1 there are patients with *GNAS*wt, DG tumors (mainly pure somatroph PitNETs but also tumors that are positive for both GH and LH) with low frequency of invasive growth. These tumors are positive for both *NR5A1* (SF-1) and *GIPR* expression and probably include the patients with paradoxical GH response to glucose intake, according to previous studies [11]. These patients are the least frequent and they account for approximately 20% of patients suffering from acromegaly. Transcriptomic group 2 seems to be the most numerous (46% of acromegaly patients). It includes mostly *GNAS*mut, densely granulated tumors as determined with electron microscopy, diagnosed generally as DG somatotroph PitNETs, and additionally mixed GH/PRL tumors. The smaller tumor size found in this group is concordant with the observation that a favorable gene expression profile was observed in these tumors. Accordingly, a low rate of invasive growth was reported for these patients. This corresponds to general observation of better prognosis in patients with densely granulated tumors rather than sparsely granulated tumors [5]. In the third transcriptomic group there are sparsely granulated tumors with a low frequency of *GNAS* mutations diagnosed mainly as SG somatotroph PitNETs. They account for approximately 35% of tumors causing acromegaly. These tumors have the highest rate of invasive growth and unfavorable gene expression profile (low expression of *CDH1*, *RORC*, and *ESRP1* and high *ARRB1*, *MKI67*, *ZEB1*, *STAT3,* and *TGFB1* levels). They also have a high level of *DRD1* gene expression, encoding stimulatory dopamine receptor that activates cAMP pathway [48]. This may affect the results of treating these patients with dopamine analogs like cabergoline. In general, the identified gene expression patterns correspond to literature data indicating that patients with sparsely granulated tumors are considered to have high risk tumors [5] as they have worse prognosis, lower rate of postoperative remission, and tumors with frequent invasive growth, a tendency to regrow after surgery, and lower response to SRLs [5]. Our transcriptomic data indicate that these tumors may be significantly driven by ghrelin signaling. They have higher expression of ghrelin receptor gene as compared to densely granulated tumors. The role of ghrelin signaling in GH secretion and pathogenesis of somatotroph PitNETs was already determined [49]. Importantly, the ghrelin receptor was recognized as a therapeutic target [50]. Small inhibitors against this receptor are available and were already subjected to clinical trials on treatment of various human diseases [50]. The use of this therapeutical approach may be potentially beneficial in sparsely granulated tumors with high *GHSR* expression and it could complement standard treatment with SRLs.

## 5. Conclusions

Whole-transcriptome analysis revealed three distinct molecular subtypes of somatotroph PitNETs. Each tumor subtype has a distinct molecular profile including gene expression pattern and frequency of *GNAS* mutations as well as profile of diagnostic histological features based on results of immunohistochemical staining and electron microscopy.

## Figures and Tables

**Figure 1 cells-11-03846-f001:**
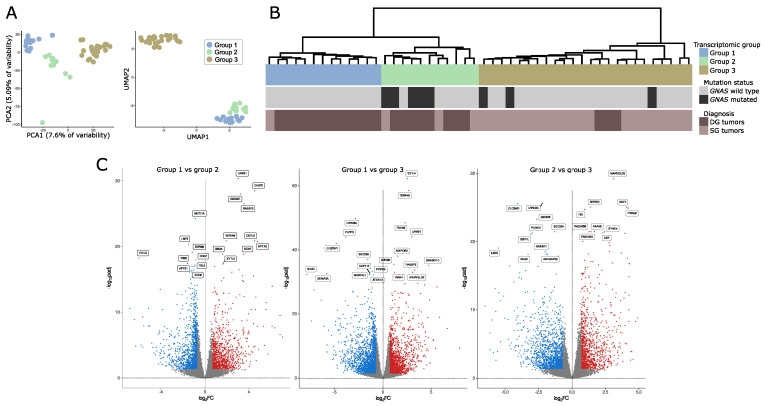
Gene expression in somatotroph tumors. (**A**) Principal component analysis (PCA) and uniform manifold approximation and projection (UMAP) results based on the expression data for the entire set of genes that indicate the presence of three transcriptomic groups of somatotroph tumors. (**B**) Hierarchical clustering of somatotroph PitNETs according to the expression data for the entire set of genes, presented with basic diagnostic data and *GNAS* mutation status. DG stands for densely granulated somatotroph PitNET, while SG stands for sparsely granulated somatotroph PitNET. (**C**) Comparison of genes expression in pairs of particular transcriptomic groups.

**Figure 2 cells-11-03846-f002:**
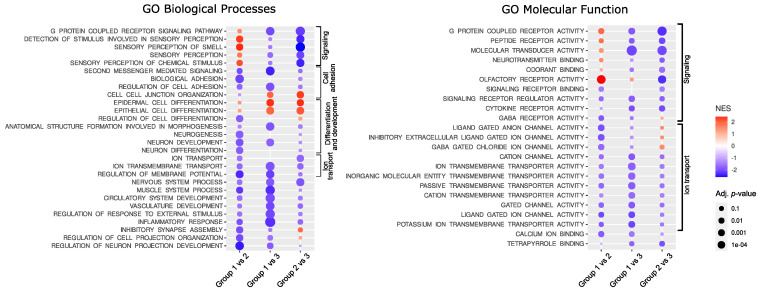
Results of Gene Set Enrichment analysis. NES—normalized enrichment score.

**Figure 3 cells-11-03846-f003:**
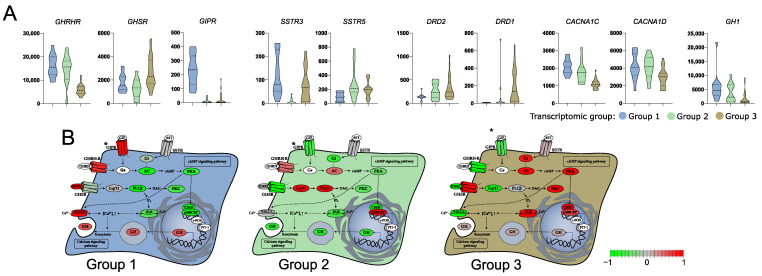
Difference in the expression of genes involved in growth hormone (GH) secretion pathway in three subtypes of somatotroph tumors. (**A**) Expression levels of differentially expressed genes coding for cell surface receptors involved in GH synthesis and secretion. Distribution of normalized RNA-seq read counts is presented. (**B**) Scaled normalized RNA-seq read counts of genes encoding each element of GH synthesis pathway were visualized on KEGG pathway with pathview. Scaled expression values (normalized RNA-seq read counts) of multiple genes were presented for the pathway elements if it is composed of more than one gene, according to KEGG Pathway Database annotation. * GIPR receptor was added manually, according to literature data [25].

**Figure 4 cells-11-03846-f004:**
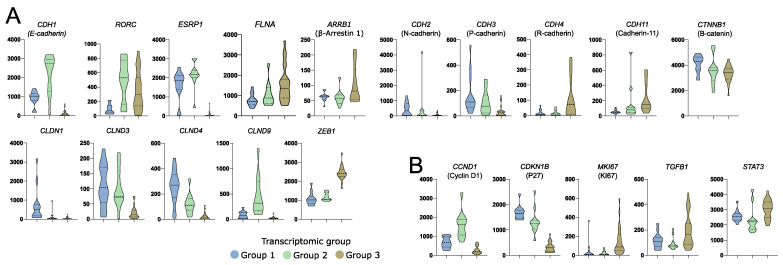
Differences in the expression of known somatotroph pituitary tumor-related genes in three subtypes of somatotroph tumors. Distribution of normalized RNA-seq read counts are presented. (**A**) Genes encoding proteins involved in epithelial-mesenchymal transition. (**B**) Genes related to somatotroph pituitary tumor growth.

**Figure 5 cells-11-03846-f005:**
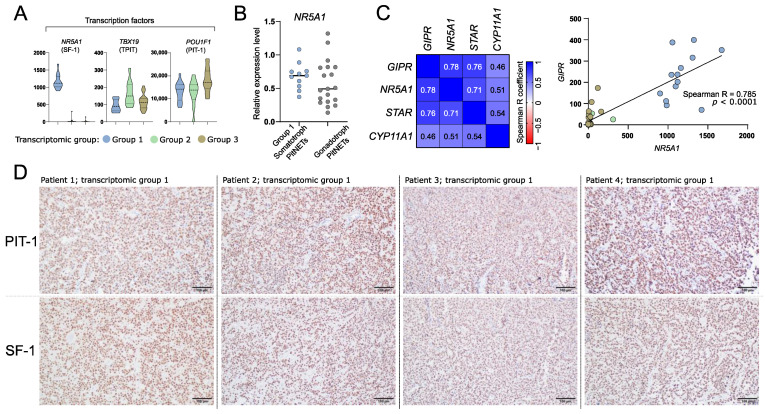
The expression of differentially expressed genes (DEGs) that are related to identity of anterior pituitary cells. (**A**) The expression levels of transcription factors specific for anterior pituitary lineages. (**B**) The comparison of *NR5A1* (SF-1) expression in somatotroph and gonadotroph PitNETs based on qRT-PCR measurement. (**C**) Co-expression of *GIPR* and steroidogenesis-related genes that were found as differentially expressed in comparison of tumors from transcriptional group 1 with groups 2 and 3. Normalized RNA-seq read counts were analyzed. (**D**) Representative examples of immunohistochemical staining of somatotroph PitNETs of transcriptomic group 1 (tumors with high *GIPR* and *NR5A1* expression) with antibodies against PIT-1 (upper panel) and SF-1 (lower panel); magnification ×200.

**Figure 6 cells-11-03846-f006:**
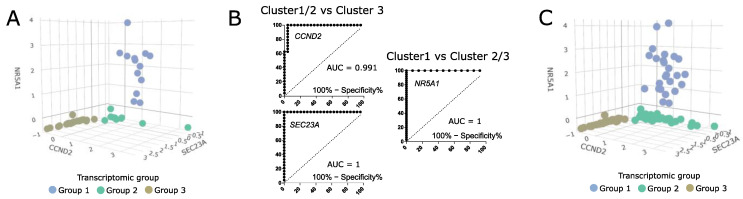
The value of classification based on three expression markers *NR5A1*, *CCND2,* and *SEC23A*. (**A**) The expression levels of marker genes in 48 samples categorized based on RNA-seq data. Scaled qRT-PCR expression values are presented. (**B**) ROC curve analysis of three selected marker genes. (**C**) The values of marker gene expression in entire patient cohort, with classification of the samples. Six unclassified tumors were excluded. Scaled qRT-PCR expression values are presented.

**Figure 7 cells-11-03846-f007:**
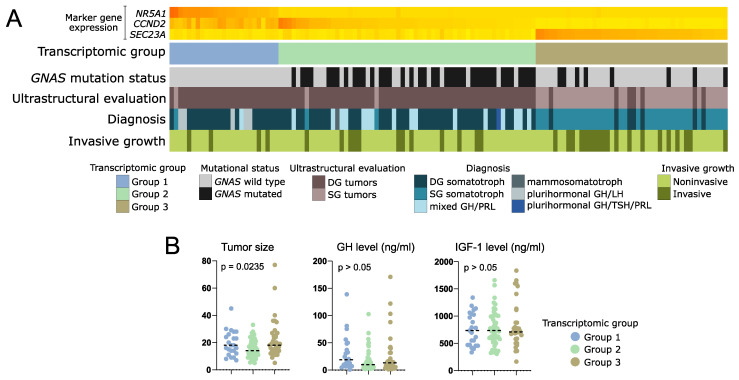
Clinical data of patients with somatotroph PitNETs categorized according to expression of marker genes into three transcriptomic groups. (**A**) Presentation of categorical patient characteristics. (**B**). Comparison of quantitative patient data including tumor size (maximal tumor diameter) as well as GH and IGF-1 plasma level. Horizontal dashed line indicates median. *p*-value assessed with Kruskal–Wallis test.

**Figure 8 cells-11-03846-f008:**
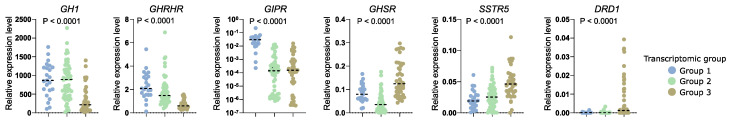
Relative expression level of key regulatory genes involved in somatotroph cell functioning measured with qRT-PCR in the entire patient cohort, with patients stratified according to three transcriptomic groups. Horizontal dashed line indicates median.

**Table 1 cells-11-03846-t001:** Summary of demographical and clinical features of patients with acromegaly.

Clinical Feature	All Acromegaly Patients	Patients Included In RNA-Seq
Number of patients	*n* = 134	*n* = 48
Sex		
Female	73/134 (54.5%)	30/48 (62.5%)
Male	61/134 (45.5%)	18/48 (37.5%)
Age at surgery (years; median (range))	44 (22–74)	39 (22–74)
GH (µg/dL; median (range))	11.8 (0.4–171) *	9.77 (0.89–177)
IGF-1 (µg/dL; median (range))	735 (166–1836) **	698 (166–1600)
Tumor size—max. diameter (mm; median (range))	16 (5–77)	18 (5.1–77)
Invasive tumor growth		
Invasive tumors (Knosp grade III, IV)	39/134 (29.1%)	22/48 (45.8%)
Noninvasive tumors (Knosp grade 0, I, II)	95/134 (70.9%)	26/48 (54.2%)
Ultrastructural evaluation		
Sparsely granulated (SG tumors)	43/134 (32%)	24/48 (50%)
Densely granulated (DG tumors)	91/134 (67.9%)	24/48 (50%)
Diagnosis		
DG somatotroph tumors	66/134 (49.25%)	
SG somatotroph tumors	43/134 (31.3%)	24/48 (50%)
Mixed GH/PRL tumors	18/134 (11.9%)	24/48 (50%)
Mammosomatotroph tumors (GH/PRL)	1/134 (0.75%)	-
Plurihormonal tumors (GH/PRL/TSH)	1/134 (0.75%)	-
Plurihormonal tumors (GH/LH)	5/134 (3.7%)	-

* The exact value was not available for 23 patients; ** The exact value was not available for 25 patients.

## Data Availability

RNA-Seq data is available at ArrayExpress: E-MTAB-11889 (https://www.ebi.ac.uk/arrayexpress/experiments/E-MTAB-11889; access date 1 December 2022).

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
