# Peer review of "Transcriptomic Classification of Pituitary Neuroendocrine Tumors Causing Acromegaly"

_cells, 2022, doi:10.3390/cells11233846_

Round 1

Reviewer 1 Report

The present study investigated transcritpomic profiles in various somatotroph PitnETs and three distinct molecular subtypes were identified. These data would be valuable for designing therapeutic strategies for PitNETs. The following are some concerns with this study.

1.What is the major difference among three groups? The conclusive description for this problem seems lacking.

2. Fgure 2, NES-normalized enrichment score needed modification to more suitable for the figure.

3. Figure 3, according to figure 3a, gh expression pattern: group1>group2>group3, while according to figure 3b, gh expression pattern: group1>group3>group2. How to explain the contradictory result?

4. Line 66 whichencodes >>which encodes.

5. Line 115 DNA was PCR amplified is suggested to revise.

Reviewer 2 Report

The authors conducted a study to investigate whether a transcriptomic profile corresponds to the current histological/clinical classification of GH secreting pituitary neuroendocrine tumors (GH-pitnets). The study is very interesting, well written, with a strong methodology. Some comments should be addressed:

1)      All patients were previously treated with somatostatin receptor ligands, which may have altered the expression of some proteins, specially SST2, as it has been demonstrated in some studies.

2)      If all patients were treated with SRL, why not compare response to treatment among the 3 groups?

3)      I agree with the authors that the high expression levels of NR5A1 do not seem to be related to normal pituitary tissue contamination. However, the WHO classification that includes the transcription factors in the characterization of pitnets considers immunohistochemical detection of these factors, not PCR. It would be important to evaluate whether SF1 is also detected at protein level in these tumors expressing it in the RNA level.

4)      Moreover, TPIT was also detected in the three groups, although in much lower level.

5)      It would be interesting, even as a supplementary table, to visualize the clinical differences among the three groups.

Some minor comments:

1)      There are two figures 7

2)      Page 12 line 478: dopamine agonists

Round 2

Reviewer 1 Report

The opinions have been incorporated in the revised version. The manuscript could be considered for the acception.